

# Unweighting multijet event generation using factorisation-aware neural networks

Timo Janßen[1], Daniel Maître[2], Steffen Schumann[1], Frank Siegert[3] and Henry Truong[2]

**1** Institut für Theoretische Physik, Georg-August-Universität Göttingen, Göttingen, Germany
**2** Institute for Particle Physics Phenomenology, Department of Physics, Durham University, United Kingdom
**3** Institut für Kern- und Teilchenphysik, TU Dresden, Dresden, Germany

## Abstract

In this article we combine a recently proposed method for factorisation-aware matrix element surrogates with an unbiased unweighting algorithm. We show that employing a sophisticated neural network emulation of QCD multijet matrix elements based on dipole factorisation can lead to a drastic acceleration of unweighted event generation. We train neural networks for a selection of partonic channels contributing at the tree-level to $Z + 4, 5$ jets and $t\bar{t} + 3, 4$ jets production at the LHC which necessitates a generalisation of the dipole emulation model to include initial state partons as well as massive final state quarks. We also present first steps towards the emulation of colour-sampled amplitudes. We incorporate these emulations as fast and accurate surrogates in a two-stage rejection sampling algorithm within the SHERPA Monte Carlo that yields unbiased unweighted events suitable for phenomenological analyses and post-processing in experimental workflows, e.g. as input to a time-consuming detector simulation. For the computational cost of unweighted events we achieve a reduction by factors between 16 and 350 for the considered channels.

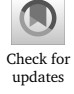

# 1 Introduction

Physics simulations for current and future high-energy accelerator experiments pose a severe computational challenge not only due to the complexity of the studied signatures and the demand for higher theoretical accuracy, but also owing to the sheer number of simulated events needed to match the enormous collider luminosities. This has sparked a wide range of algorithmic developments to accelerate key elements of the simulation tool chain and to improve their computational efficiency and thus reduce their resource requirements. Machine learning based methods play a prominent role in these developments [3].

Traditionally, the largest fraction of resources has been spent on the complex simulation of the detector response to collision final states, while the generation of the collision events constituted only $\mathcal{O}(10\% - 20\%)$ of the budget. With many recent activities reducing the computational footprint of detector simulations, the event generation speed has become a more and more important area to facilitate the full exploitation of future collider data.

This situation is amplified with the upcoming increased luminosity at the LHC and its focus turning to more complex processes. With the advent of matching and merging techniques theoretical predictions of increased precision have become accessible for these high multiplicity final states. But as the computational cost increases strongly with the multiplicity the resources needed for the theoretical description of processes of interest has surged, see for example [4]. Besides conventional approaches for improving the performance of Monte Carlo methods in event generators, more recently also machine learning methods are explored [5].

A particular challenge is related to the generation of the hard scattering component that forms the core of the event evolution, thereby representing the parton-level truth signal and/or background hypotheses in physics analyses [6,7]. In Monte Carlo event generators the hard process is assumed to be factorised from parton showers in the initial and final state as well as non-perturbative phenomena such as hadronisation and the underlying event. Algorithmically the generation of hard scattering events corresponds to the evaluation, *i.e.* the stochastic sampling, of the phase space integral over the squared transition matrix element of the considered process at a given order in perturbation theory.

There has been renewed interest in improving phase space integration and sampling techniques, mostly based on neural networks [8–14], using a variety of methods, but also Nested Sampling [15], and mixed-kernel Markov chain algorithms [16] have been investigated. Broadly speaking these approaches share the goal of adjusting a sampling distribution as closely as possible to the true target distribution, *i.e.* the actual transition matrix element. In the case of a traditional Monte Carlo integration this will typically result in events with weights of ideally small spread. However, for a resource efficient generation process for physics analyses, in particular due to very time-consuming components such as a detector simulation, events with (largely) unit weight are desirable. This is typically solved via von Neumann rejection sampling, *i.e.* an accept–reject procedure for weighted event samples. However, even with advanced adaptive sampling techniques in particular for multi-particle final states the efficiency of the unweighting procedure can be quite small, resulting in the repeated trial evaluation of

the scattering matrix element that ultimately get rejected. For LHC key processes such as the multijet-associated production of gauge bosons this might result in $\mathcal{O}(10^5)$ evaluations of the computationally expensive matrix element for a single unit-weight event [17].

This suggests a complementary opportunity for saving resources, namely the usage of a fast *and* accurate surrogate for the trial weights. A corresponding two-stage unweighting procedure that fully corrects for the potential mismatch between the surrogate weight and the actual value of the full matrix element has recently been presented in Ref. [2]. To demonstrate the algorithm, a rather simple neural network designed to replicate the weight of partonic events, represented by their external momenta was used. For tree-level contributions to $Z/W + 4$ jets and $t\bar{t} + 3$ jets production at the LHC significant gains have been observed, however, for less complex partonic channels ordinary unweighting could not be outperformed. Over the last few years more sophisticated matrix element surrogates based on neural networks have been developed [1, 18, 19], addressing tree-level and one-loop amplitudes. In particular, Ref. [1] presented a method to emulate scattering matrix elements employing the factorisation properties of QCD amplitudes in the soft- and collinear limits.

In this work we explore the potential of a combination of the approaches in Refs. [2] and [1] in unweighted event generation for multijet production processes at the LHC. To this end we generalise the method presented in Ref. [1] to the case of colour-charged initial states and massive final-state partons. We also explore, for the first time, the emulation of colour-sampled QCD amplitudes in the colour-flow decomposition. With an implementation in the SHERPA event generator framework [20,21] we benchmark tree-level contributions to $Z + 4, 5$ jets and $t\bar{t} + 3, 4$ jets production. The paper is structured as follows: In Sec. 2 we review the dipole emulation model of Ref. [1] and present our new developments to address hadronic collisions and massive final-state partons. In Sec. 3 we review the unweighting procedure worked out in Ref. [2]. In Sec. 4 we discuss the implementation of both algorithms in the SHERPA framework, and present our results obtained for selected partonic channels contributing to $pp \to Z + 4, 5$ jets and $pp \to t\bar{t} + 3, 4$ jets.

## 2 Improved matrix element emulation using neural networks

To demonstrate the accelerated surrogate unweighting algorithm in Ref. [2] the authors used only a simple neural network as model for the event weights. It was clear from the study that a bottleneck in this procedure was the accuracy of the event weight approximations coming from the surrogate model, leading to low efficiencies in the second unweighting step. In this work we consider replacing that simple model with the factorisation-aware neural network model introduced in Ref. [1], which has been shown to exhibit more accurate predictions on a per-point basis compared to existing methods. Below we review the construction of this model and detail the necessary extensions to facilitate multijet production processes at the LHC.

### 2.1 Neural networks based on dipoles

In this section we briefly review the framework from Ref. [1], where an ansatz for matrix elements based on the factorisation properties of QCD matrix elements in their soft and collinear limits was used. This factorisation can be depicted as

$$|\mathcal{M}_{n+1}|^2 \to |\mathcal{M}_n|^2 \otimes \mathbf{V}_{ijk}\,, \tag{1}$$

where the matrix element in the $(n + 1)$-body phase space reduces to a matrix element in the momentum mapped $n$-body phase space multiplied by a singular factor, $\mathbf{V}_{ijk}$. For single infrared limits, $\mathbf{V}_{ijk}$ contains all the singularity structure of the matrix element. In the dipole

formalism (originally presented by Catani and Seymour for massless partons in [22] and later generalised by Catani, Dittmaier, Seymour and Trócsányi to account for finite masses in [23]) these singular factors are encapsulated in the dipole functions $D_{ijk}$. This factorisation property of matrix elements leads to the form of our ansatz, which is inspired by the dipole factorisation formula. It is given by

$$\left|\mathcal{M}_{n+1}\right|^2 \simeq \sum_{\{ijk\}} C_{ijk} D_{ijk}, \tag{2}$$

where $i$, $j$, and $k$ denote the three partons involved in the dipole function. Instead of fitting the matrix element directly where there are divergences in the infrared regions of phase space, we let the neural network fit the coefficients $C_{ijk}$ as a function of phase space. These coefficients are more well-behaved than the full matrix element in the soft and collinear limits as the singular behaviour is described by the dipole functions. By combining these well-behaved coefficients with the analytically known dipoles, we produce an approximation for the matrix element. This enables the fitting of matrix elements across the entire sampled phase space with a single neural network. Whilst the model predicts the $C_{ijk}$ coefficients, it should be noted that these are not entirely meaningful by themselves. Only once they are combined with the corresponding dipoles do we get the approximation of the matrix element. It is this approximation of the matrix element which should be seen as the model prediction and which appears in the loss function.

In singly unresolved limits, only relevant dipoles are large and so constrain the corresponding $C_{ijk}$ in the fit. Outside of these limits all dipoles are of similar order of magnitude and the ansatz in Eq. (2) is more under-constrained. In these regions of phase space, the excellent fitting capabilities of neural networks are leveraged to interpolate the non-singular matrix elements. The accuracy achieved in this approach is due to the fact that the coefficients being fit by the network, for single soft or collinear kinematics, are free of divergences. This facet of the emulation model makes it particularly apt for the case of multijet production processes where matrix elements are plagued with many well-understood divergent structures. In Sec. 4 we will apply the method to emulate jet-production processes at the tree level, where the fiducial phase space is constrained by jet cuts, *i.e.* a minimal separation of all QCD parton pairs and a moderate transverse momentum threshold.

We note that Eq. (1) is not to be interpreted as a recursion relation, instead it depicts the isolation of a single infrared limit into the dipole $D_{ijk}$ leaving the coefficients $C_{ijk}$ to capture the behaviour of the non-divergent $n$-body matrix element. In principle, this model could be extended to cases of multiple unresolved partons where the divergences are captured by multiple $D_{ijk}$ functions, again leaving $C_{ijk}$ well-behaved. This was examined briefly in Section 3.3 of Ref. [1] where the model performance was tested on more complex infrared configurations.

In this work we employ networks of similar size and complexity to those in Ref. [2], namely, we use KERAS [24] and TENSORFLOW [25] to build a neural network (NN) model with four hidden layers, each consisting of 128 nodes. These hidden layers use the swish activation function [26, 27] and their weights are initialised according to the Glorot Uniform distribution [28], which aims to keep the variance of activations similar across all hidden layers in order to prevent exploding or vanishing gradients during network training.

The swish activation function, $\frac{x}{1+\exp(-x)}$, has a similar shape to the more well-known ReLU activation function, but it is a smooth continuous function allowing small negative values, instead of thresholding them to 0 like ReLU does. We find that in practice swish outperforms ReLU in all our trained models. We also find that instead of using a linear activation function in the output layer, using a swish activation function is more performant.

The NN is fitted to the data generated from SHERPA (see Section 4.2 for more information on the generation of data) by minimising the loss function encoding the discrepancy between the prediction made with the neural network to the true matrix element provided by SHERPA.

Table 1: Hyperparameters of the neural network and their values.

| Parameter | Value |
|---|---|
| Hidden layers | 4 |
| Nodes in hidden layers | 128 |
| Activation function | swish [26, 27] |
| Weight initialiser | Glorot uniform [28] |
| Loss function | MSE |
| Batch size | 512 |
| Optimiser | ADAM [29] |
| Initial learning rate | $10^{-3}$ |
| Callbacks | EarlyStopping, ReduceLROnPlateau |

We use the mean squared error (MSE) as the loss function, with the training optimised using ADAM [29] with an initial learning rate of $10^{-3}$. The learning rate is reduced when the validation loss shows no improvement for 30 epochs of training by using the `ReduceLROnPlateau` callback, and `EarlyStopping` is used to terminate training when there is no improvement in the validation loss after 60 epochs. A summary of the neural network hyperparameters is given in Tab. 1 for reference.

As inputs to our generalised network model we feed: the 4-momenta of all initial- and final-state particles, the phase-space mapping variables corresponding to the dipoles in the ansatz, denoted as $y_{ijk}$[1], and the kinematic invariants $s_{ij}$ for all pairs of particles in the process considered. Dipoles can be classified depending on whether the emitter and spectator are in the initial- or final-state, and whether they are massless or massive. The four classes of dipoles are FF (final-state emitter, final-state spectator), FI (final-state emitter, initial-state spectator), IF (initial-state emitter, final-state spectator), and II (initial-state emitter, initial-state spectator), where each class can be massless or massive. Each of these dipole configurations have a corresponding phase-space mapping. To illustrate the general form of these mappings, we quote the massless FF case here,

$$y_{ijk} = \frac{p_i p_j}{p_i p_j + p_j p_k + p_i p_k} \, ,$$ 
(3)

where it is understood that the other mappings are also functions of the external momenta (and masses if present). Note that the $y_{ijk}$ and the $s_{ij}$ input variables are *not* independent from the external momenta. To aid the neural network training process, we pre-process the $y_{ijk}$, such that the shape and widths of their distributions are similar for the different dipole configurations. This amounts to

$$y_{ijk} \rightarrow \begin{cases} \log(1 - y_{ijk}), & \text{if massless FI, IF, or II dipole}, \\ \log(y_{ijk}), & \text{otherwise}. \end{cases}$$ 
(4)

The kinematic invariants are also transformed with the logarithm $s_{ij} \rightarrow \log(s_{ij})$ as they can span many orders of magnitude. It should be noted that the particles involved in the dipole functions and mapping variables denoted by the subscript $ijk$ are the colour-charged particles in the initial- and final-state. Non colour-charged particles, for example, electrons and

---

[1]The initial-state phase-space mapping variables are referred to as $x_{ijk}$ in [22, 23] but we will refer to all phase-space mappings as $y_{ijk}$ for brevity.

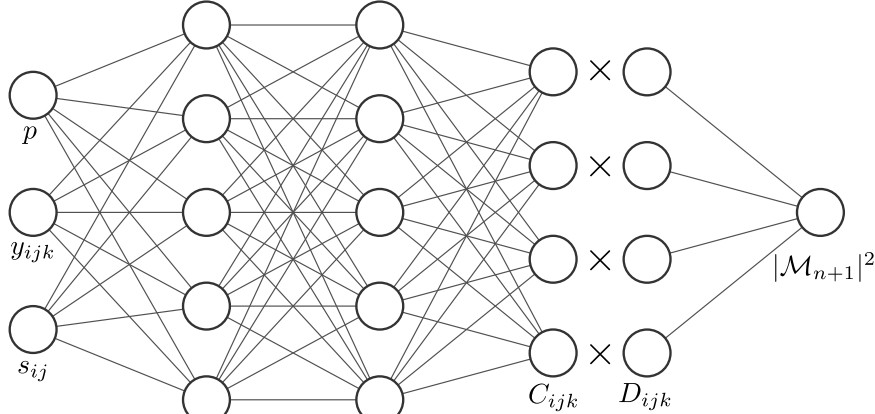

Figure 1: A simplified sketch of our neural network emulator showing inputs, hidden layers, and outputs $C_{ijk}$.

positrons, do not appear in the dipoles, but nevertheless their momenta and kinematic invariants are fed into the network as inputs such that it learns of their dependence. All of these inputs are standardised to zero mean and unit variance, with the 4-momenta being standardised along each component.

To use them more effectively in the loss function, we pre-process the matrix elements as

$$|\mathcal{M}_{n+1}|^2 \to \mathrm{arsinh}\left(\frac{|\mathcal{M}_{n+1}|^2}{S_{\mathrm{pred}}}\right), \tag{5}$$

and standardise to zero mean and unit variance. $S_{\mathrm{pred}}$ is the prediction scale taken to be the minimum matrix element value found in our training set. This transformation aids the neural network in training by reducing the span of the target distribution.

The output nodes of our neural network correspond not to the matrix elements directly, but instead to the dipole coefficients. The raw outputs, denoted by $c_{ijk}$, are transformed to the coefficients appearing in Eq. (2), $C_{ijk}$, via the transformation

$$C_{ijk} = S_{\mathrm{coef}} \times \sinh(c_{ijk}), \tag{6}$$

where $S_{\mathrm{coef}}$ is the coefficient scale, taken to be $S_{\mathrm{pred}}/S_{\mathrm{dipole}}$. $S_{\mathrm{dipole}}$ is the representative value of a dipole, which we take to be the median of all dipoles in our training set. The neural network prediction is made by using Eq. (2) to combine the predicted $C_{ijk}$ coefficients with the corresponding dipoles $D_{ijk}$. In order to compare with the scaled target matrix elements, we have to transform the neural network predicted matrix element with Eq. (5) with the same $S_{\mathrm{pred}}$. We can then compare the matrix element as predicted by the neural network, with the truth value, as given by SHERPA, in the MSE loss function. A diagram illustrating the NN emulator architecture is given in Fig. 1.

In Ref. [1], the neural network predictions were given by the average over an ensemble of 20 independent replicas trained on different shuffled subsets of the training set and with different initial random seeds for model weight initialisation. Here we take a similar approach by training a set of 10 replica models, however, for predictions we select the model with the lowest validation loss. We stress that this is not a special choice as all individual replica models converge to a similar point. As an illustrative example, we plot in Fig. 2 the loss curves for the partonic channel $gg \to e^-e^+ggd\bar{d}$, which is a leading-order contribution to $Z + 4$ jets production at the LHC. We observe convergence across all replicas with training terminating at similar values of the MSE.

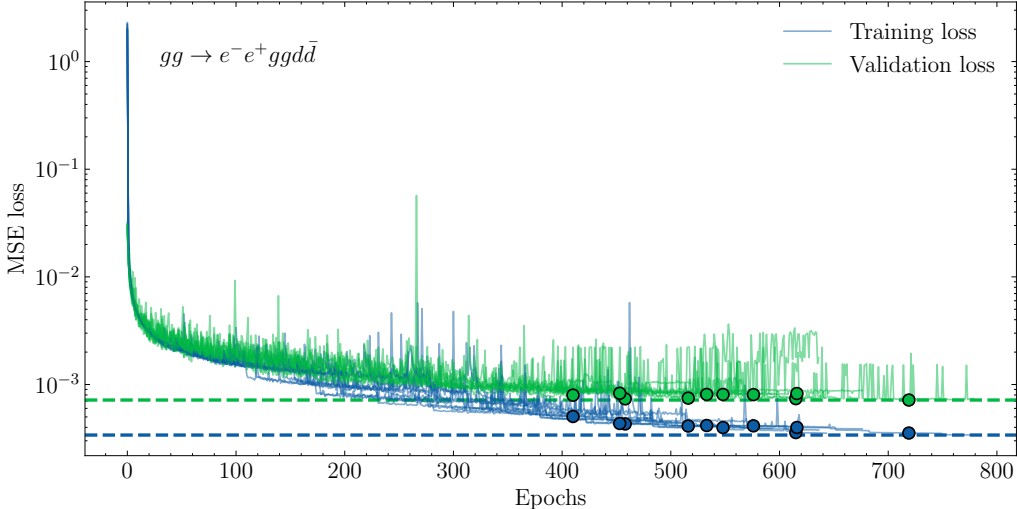

Figure 2: Training and validation loss recorded during training for 10 replica models, for the $gg \rightarrow e^- e^+ gg d\bar{d}$ channel, shown as solid lines. The MSE loss is the mean squared difference between the transformed predictions and transformed truth values. The epochs at which training is terminated are illustrated as the solid circles. We depict the training and validation loss of the selected model in dashed horizontal lines.

The reasoning behind ensembling a prediction is to reduce the effects of stochasticity of the training process, to reduce random model weight initialisation, and to reduce variance in the prediction. In this work we strive for a balance of accuracy and speed, meaning it is advantageous to use a single model to make predictions. The reasoning is as follows. In an ensemble of models where replicas are trained on different subsets of the same training data, there is overlapping information learnt by the individual models. This leads to diminishing returns in predictive accuracy, meaning that whilst evaluation time grows linearly with the number of replicas, accuracy does not. We have therefore observed a single model to be the most performant configuration. It is important to stress that this does not mean that one model cannot be sufficiently accurate, as we will demonstrate in Sec. 4.2.

This decision to use only a single NN for predictions also guided our choice of number of nodes in the hidden layers. With 128 nodes we reach a balance of having enough parameters to model the matrix elements whilst reducing the effects of overfitting. Decreasing the number of nodes in the hidden layers to create a more compact NN has little effect on the evaluation time for a single network when we use the ONNX Runtime [30] for evaluation, there would only be loss in accuracy which represents a decrease in overall unweighting efficiency.

## 2.2 Extension to initial-state and massive partons

In Ref. [1], the authors considered jet production processes initiated via electron–positron annihilation where only final-state QCD radiation occurs, meaning the set of dipoles built into the emulation model were of the FF kind. Furthermore, the model was restricted to the production of massless QCD partons.

In this work we consider the extension to hadronic initial states, which is relatively straightforward: we need to account for the additional radiation that comes from the colour-charged initial-state particles. To this end, we add the initial-state dipoles to the ansatz, namely, we add the IF, FI and II splitting configurations. This means that the emitter $i$, and spectator $k$, in the ansatz can now be in the initial-state. Illustrations for the complete set of dipoles now

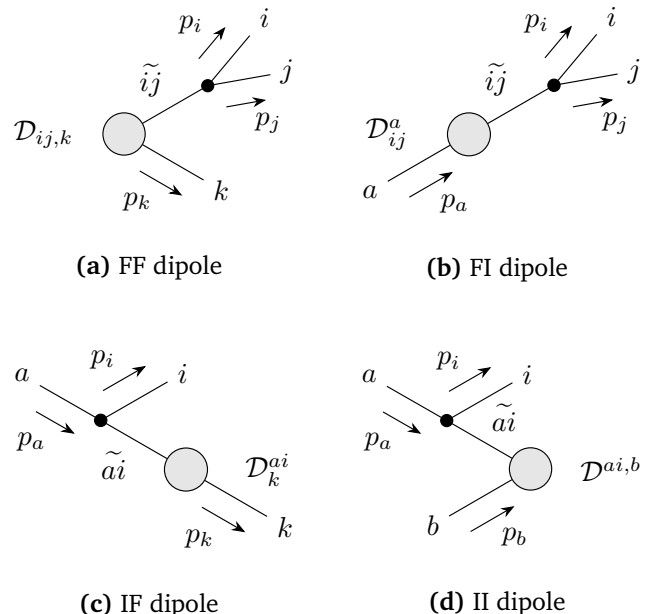

**(a)** FF dipole  **(b)** FI dipole

**(c)** IF dipole  **(d)** II dipole

Figure 3: Schematic diagrams of the four classes of dipoles. The dipoles are named according to whether the emitter and spectator are in the initial (upper indices) or final state (lower indices). Each dipole consists of a composite particle (denoted by tilde) that decays into two partons, and a spectator that recoils to conserve momentum. The grey blob represents the hard scattering process, with incoming and outgoing lines representing initial- and final-state partons, respectively. The black circle represents the splitting function within the dipole function which contains the divergent behaviour.

included in the model are shown in Fig. 3.

To showcase the extension to massless initial state dipoles we consider the emulation of tree-level matrix elements for the partonic channels $gg \rightarrow e^-e^+ggd\bar{d}$ and $gg \rightarrow e^-e^+gggd\bar{d}$, which are leading order contributions to $Z + 4$ jets and $Z + 5$ jets production at the LHC, respectively. As validation of the emulation accuracy of the NN model for this extension to initial states, we examine the ability of the model to predict matrix elements across the sampled phase space, but in particular for the case of soft and collinear kinematics, where QCD matrix elements are strongly enhanced. We plot in Fig. 4 a 2d histogram of the truth-to-prediction ratio, $|\mathcal{M}|^2_{\text{true}}/|\mathcal{M}|^2_{\text{pred}}$, against the true value, $|\mathcal{M}|^2_{\text{true}}$, for 1M $gg \rightarrow e^-e^+ggd\bar{d}$ test events with standard cuts as described in Sec. 4.2. Along the sides, we plot the marginal distributions of the matrix element (top) and the ratio (right). The results illustrate that the ratio depicting model accuracy is centred around the ideal value of 1, with a steep drop off. This applies to the bulk of the events, as depicted by yellow coloured bins, tightly constrained to a narrow band. The purple coloured bins represent low population bins, or single points, which shows that the tails of the ratio distribution are primarily seen for smaller matrix element weights. Furthermore, the model accuracy remains high for the largest values of the matrix element, signalling that the infrared behaviour is well controlled. This is a key property of the factorisation-aware model. The emulation performance for the $Z + 5$ jets process is presented in Sec. 4.2.

An additional extension we study in this article is the inclusion of massive dipoles to our ansatz. This allows us to examine QCD processes with massive partons which is of particular importance for top-quark pair production in association with jets. We include the massive FF, FI, and IF dipoles from Ref. [23] into the emulation model. The massive dipoles are generalisa-

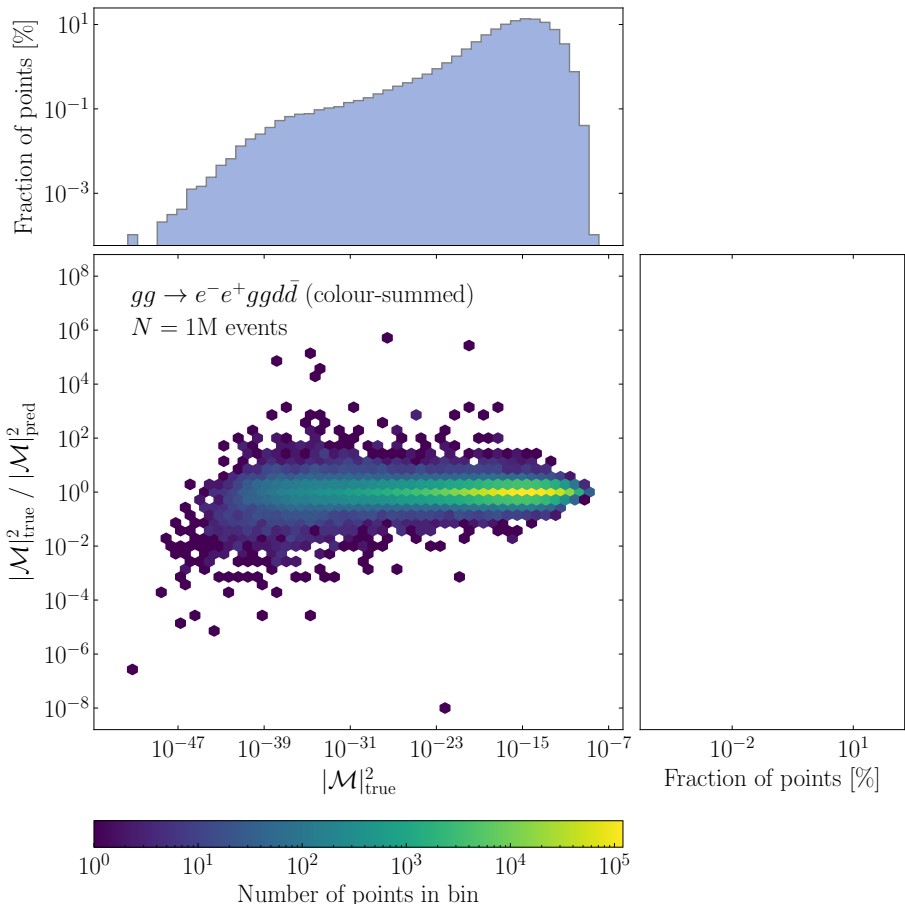

Figure 4: 2d histogram showing the distribution of truth-to-prediction ratios of the matrix element against the value of the true matrix element for the $Z + 4j$ process $gg \rightarrow e^-e^+ggd\bar{d}$. Along the axes, we plot the marginal distributions of the matrix element (top), and the truth-to-prediction ratio (right). High population bins are illustrated as yellow, with low population bins, down to single points, are depicted in purple.

tions of the massless dipoles, meaning in principle it would be possible to remove the massless dipoles from the ansatz. However, in practice, we only include the minimal set of necessary dipoles for a given partonic channel and so the inclusion of the massless dipoles reduces overall computational cost due to their relatively simpler expressions. With the massive dipoles implemented, our model contains the complete set of dipoles and is in principle able to take advantage of the factorisation-aware model for arbitrary processes involving QCD-enhanced behaviour at tree-level.

In order to showcase the extension to massive dipoles, we consider emulating tree-level matrix elements of three partonic channels: $gg \rightarrow t\bar{t}ggg$, and $u\bar{u} \rightarrow t\bar{t}gd\bar{d}$, contributing to leading order $t\bar{t} + 3$ jets production, and $ug \rightarrow t\bar{t}gggu$ which is a leading order contribution to $t\bar{t} + 4$ jets production. To validate the inclusion of these massive dipoles into the model, we show in Fig. 5 the deviation similar to Fig. 4 but for 1M tree-level events of $gg \rightarrow t\bar{t}ggg$ in proton–proton collisions at $\sqrt{s} = 13$ TeV, with cuts described in Sec. 4.2. We again observe the narrow yellow band, indicating that the bulk of the test events are accurately predicted, with the outliers corresponding to smaller matrix element values. The infrared behaviour is well captured by the model as can be seen by the narrow head for the largest matrix element values. For emulation performance of channels not described here, we refer the reader to Sec. 4.2 and

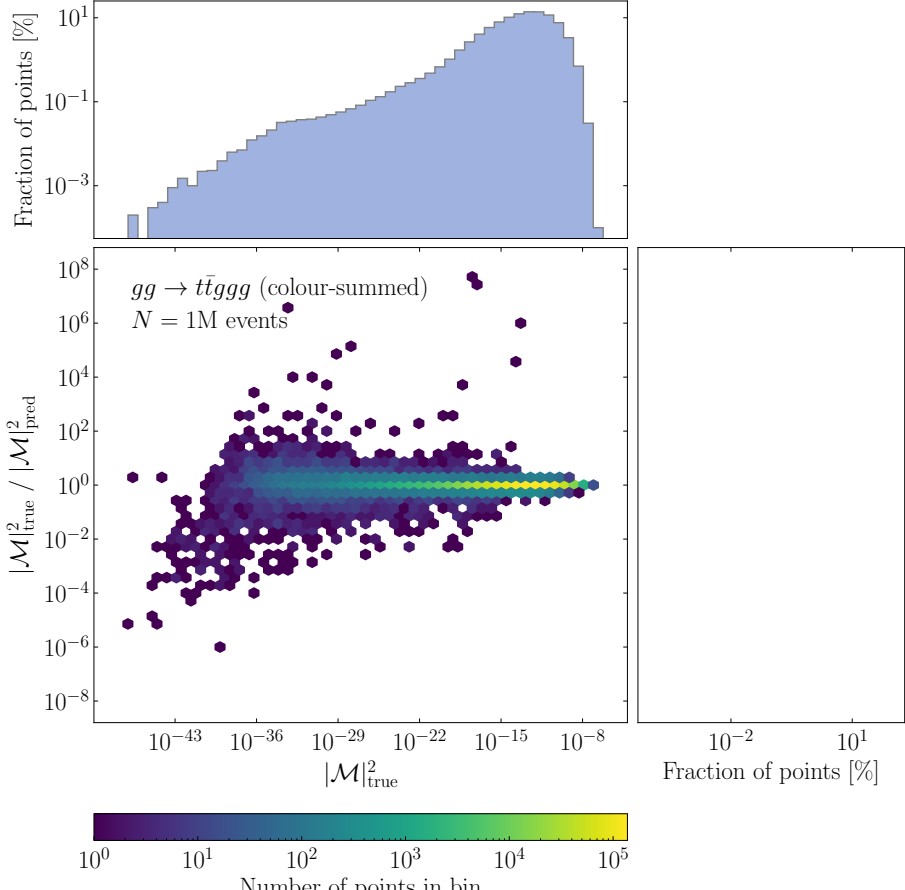

Figure 5: 2d histogram showing the distribution of truth-to-prediction ratios of the matrix element against the value of the true matrix element for the $t\bar{t} + 3j$ process $gg \to t\bar{t}ggg$. Along the axes, we plot the marginal distributions of the matrix element (top), and the truth-to-prediction ratio (right).

App. A.

## 2.3 Colour-sampled matrix elements

The discussion so far has been focused on the emulation of colour-summed matrix elements as it was the case for Ref. [1]. In this work we take the first steps towards emulating colour-sampled matrix elements, such as those obtained from the COMIX generator [31,32].

Based on the colour-flow decomposition of QCD amplitudes [33,34], for each event, the generator samples a momentum configuration *and* a valid colour assignment, *i.e.* colour indices. The colour assignment thereby is represented by a vector of integers, $C$, where entries in the vector, $c_i \in \{1, 2, 3\}$, denote the colour assigned to a colour-charged parton in the process. Gluons have two colour indices corresponding to colour and anti-colour, whereas quarks/anti-quarks carry only one index.

We add this vector of colour assignments as an additional input to the NN to include the colour-sampled information from the generator. We one-hot encode the colour assignments such that colours are represented by 3-element vectors, *e.g.* $R = [0, 0, 1]$, $G = [0, 1, 0]$, and $B = [1, 0, 0]$, as the integer representation of colour assignments is not useful to the NN.

Given the actual colour of a parton is ambiguous, the matrix element should be invariant to any cyclic permutation of the specific colour assigned to a given quark or gluon. To give an ex-

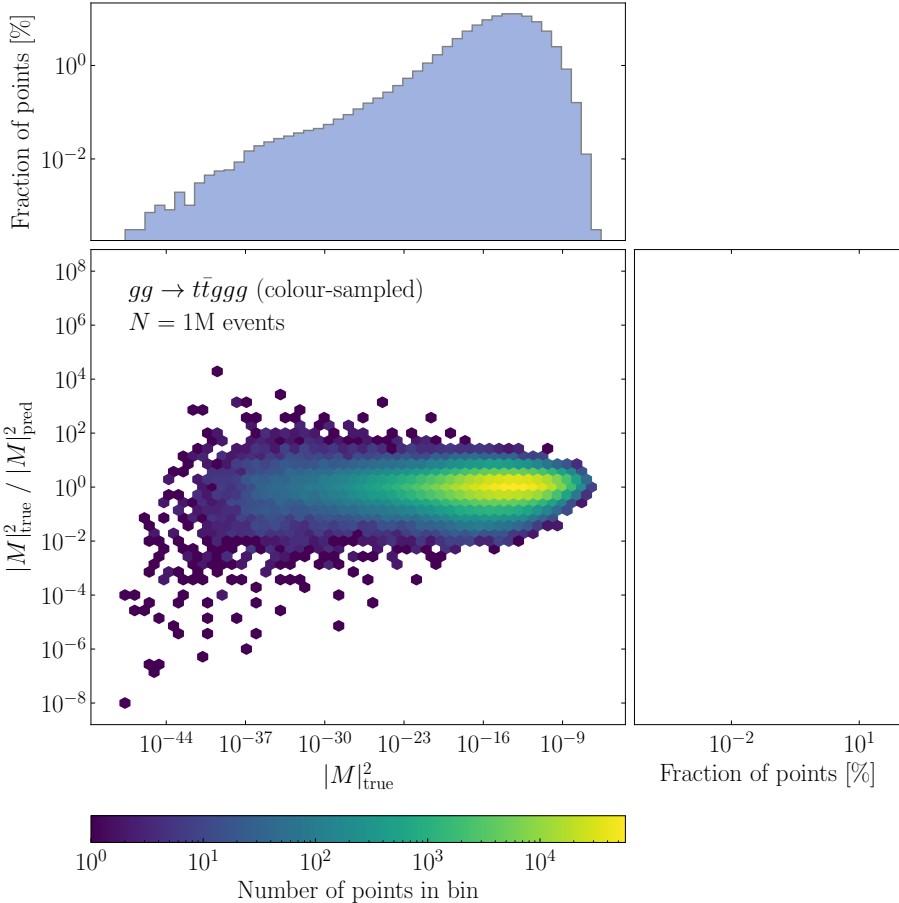

Figure 6: Truth-to-prediction ratio for colour-sampled $gg \to t\bar{t}ggg$ matrix elements against the colour-ordered partial amplitudes, $|M|^2$. Marginal distributions are plotted for the matrix elements (top) and ratios (right).

ample, the three permutations $C_1 = [R, B, B, G, R]$, $C_2 = [G, R, R, B, G]$, and $C_3 = [B, G, G, R, B]$ of a five colour assignment would lead to the same matrix element weight. To aid the NN in learning this behaviour, we take the three permutations and duplicate the other model inputs such that the training data is enlarged by a factor of three. This did not cause us to run into any computational bottlenecks in terms of memory or time taken to train the models. Note that this duplication of data is not required when making predictions.

The rest of the inputs to the NN model remain identical. We study to what extent a naive approach of using the same dipole functions, which are most suitable for colour-summed matrix elements, works for the case of colour-sampled matrix elements. In the future, a more promising approach might be the application of coloured dipole terms directly. Their form has already been derived [35] and implemented for the dipole subtraction in the COMIX event generator [31] but is not implemented in our NN-based model yet.

To illustrate the emulation accuracy of colour-sampled matrix elements, here denoted $|M|^2$, we plot the truth-to-prediction ratio in Fig. 6 for the $gg \to t\bar{t}ggg$ channel. While we again observe the property of well-behaved predictions for the larger matrix elements, evidently, the ratio distribution is much wider than in the colour-summed case. This decrease in accuracy directly translates to a lower expected gain factor when using this emulator as a surrogate model for event unweighting. This is discussed further in Sec. 4.2 where we elaborate on specific reasons for this decrease in accuracy and present possible future endeavours.

# 3  Event unweighting utilising matrix element surrogates

The unweighting of hard-scattering parton-level event samples constitutes an important step in the simulation of scattering events. The obtained unit-weight events then get passed on to subsequent evolution stages, including QCD parton showers, hadronisation, and possibly a detector simulation. However, the unweighting, based on rejection sampling, can pose a severe computational challenge, in particular when the evaluation time of the matrix element is long and the efficiency of the unweighting is rather low. To address this challenge Ref. [2] proposed a novel two-stage rejection sampling algorithm based on fast surrogates that we briefly review in this section. We furthermore generalise the performance measures to the case where the surrogate replaces the matrix element only, rather than its combination with the phase space weight as was the case in Ref. [2].

## 3.1  Two-stage unweighting method

The Monte Carlo method provides a numerical procedure to estimate integrals, *e.g.* partonic cross sections in high energy physics. When the integrand is non-trivial we use importance sampling to reduce the variance of the integral estimate. For a positive-definite target function $f : \Omega \subset \mathbb{R}^d \to [0, \infty)$ defined over the unit hypercube $\Omega = [0, 1]^d$ and a probability density function $g$ the Monte Carlo estimate of the integral

$$I = \int_\Omega f(u')\,\mathrm{d}u' = \int_\Omega \frac{f(v')}{g(v')}\,\mathrm{d}v', \quad \text{with} \quad \int_\Omega g(u')\,\mathrm{d}u' = 1, \tag{7}$$

is given by

$$I \approx \frac{1}{N} \sum_{i=1}^{N} \frac{f(v_i)}{g(v_i)} = \langle w \rangle_g, \tag{8}$$

with the pointwise event weight $w_i = f(v_i)/g(v_i)$. The points $v_i$ are drawn from the distribution $g$. A suitable $g$ can reduce the variance of the integral estimate and thereby increase the efficiency of the numerical integration. Finding such a function $g$ is a difficult task, though, as one needs a way to efficiently draw samples from it. For multimodal target functions it is attractive to use a multi-channel approach, where $g$ is defined by a mixture distribution. The weights of the channels can then be adapted automatically [36]. VEGAS [37] is an algorithm to automatically construct a sampling distribution $g$ by optimising the bin widths of a piecewise-constant function. It can also be used to remap a given $g$ or even the individual channels of a multi-channel distribution [38].

Besides the total integral we are typically interested in differential distributions of the points $u_i$, *i.e.* histograms of physical observables. Monte Carlo sampling produces weighted events so every entry in a histogram comes with a weight. Variance reduction methods like importance sampling also reduce the spread of weights but only a perfect sampler results in strictly uniform weights. A large weight spread is problematic when the samples are to be post-processed by detector simulations, as these are very expensive in terms of computation time per event. It is inefficient to apply them to events that yield a minuscule contribution to the total cross section. The alternative is to first impose a rejection sampling step to extract unit-weight samples. This converts a sample of $N^{\text{trials}}$ weighted events into a set of $N \le N^{\text{trials}}$ unweighted events by randomly accepting or rejecting every weighted event with the acceptance probability $w/w_{\max}$ where $w_{\max}$ is the maximal event weight. Even though the information of the rejected events is lost the overall efficiency can be significantly increased when detector simulation is more expensive than event generation.

---

**Algorithm 1:** Two-stage rejection-sampling unweighting algorithm using an event-wise weight estimate.

> **while** *true* **do**
>> generate phase-space point $u$;
>> calculate approximate event weight $s$;
>> generate uniform random number $R_1 \in [0,1)$;
>> **# first unweighting step**
>> **if** $s > R_1 \cdot w_{max}$ **then**
>>> calculate exact event weight $w$;
>>> determine ratio $x = w/s$;
>>> generate uniform random number $R_2 \in [0,1)$;
>>> **# second unweighting step**
>>> **if** $x > R_2 \cdot x_{max}$ **then**
>>>> **return** $u$ *and* $\widetilde{w} = \max(1, s/w_{max}) \cdot \max(1, x/x_{max})$
>>> **end**
>> **end**
> **end**

---

A convenient measure for the performance of a Monte Carlo event generator is the unweighting efficiency $\epsilon$ of the rejection sampling step, defined as

$$\epsilon := \frac{N}{N^{\text{trials}}} \,. \tag{9}$$

For a large number of trial events it can be estimated by

$$\epsilon \approx \frac{\langle w \rangle}{w_{\text{max}}} \,, \tag{10}$$

where $\langle w \rangle$ is the mean of the $N_{\text{trials}}$ weights in the event sample. The average number of target function evaluations needed to get one accepted event is then given by $1/\epsilon$. Similar to how the uncertainty on the integral estimate can be diminished by variance reduction methods, the unweighting efficiency can be increased by optimising the sampling density $g$ for smaller $w_{\text{max}}$.

There is another way of reducing the computational footprint especially if the target function takes a long time to evaluate and has a rather low unweighting efficiency. This is typically the case for high multiplicity scattering processes. The enormous growth in the number of contributing Feynman diagrams makes high multiplicity matrix elements increasingly expensive. At the same time, the high dimensionality of phase space renders it difficult to find a sampling density $g$ that is well adapted to the target everywhere in the integration volume. Consequently, the unweighting efficiency typically decreases with increasing multiplicity, see for example [17]. In this situation one can reduce the overall event generation time through replacing the expensive matrix element by a fast and accurate surrogate. The inaccuracy inevitably introduced in this procedure can be fully corrected for in a second unweighting step, resulting in an unbiased method [2]. An outline of the algorithm is given in Alg. 1. In addition, a more extensive explanation follows below.

We begin by generating a weighted trial event in the conventional way. In a first unweighting step we then compare the surrogate weight $s$ to the weight maximum $w_{\text{max}}$ and accept the event with probability $s/w_{\text{max}}$. For an event that gets rejected at this point we only had to evaluate the cheap surrogate. If the event gets accepted, however, we need to evaluate the true weight $w$ and attach a correction weight $x = w/s$ to the event. In a second unweighting step,

the event has an acceptance probability of $x/x_{\max}$. Like $w_{\max}$, $x_{\max}$ has to be predetermined. When the surrogate yields an accurate approximation of the true weight, a large proportion of events gets accepted in the second unweighting step. We note that the algorithm can easily be extended to the case of not strictly positive event weights as shown in [2].

Alg. 1 contains a crucial detail regarding the weight maxima, namely that even after unweighting events can end up with weights $\tilde{w} > 1$ if $s$ is larger than $w_{\max}$ or if $x$ is larger than $x_{\max}$. If the true maxima were used, this could never happen. However, given finite-sized samples an exact determination of $w_{\max}$ is realistically not possible. It is often not even desirable since a small number of points with large weights can induce a prohibitively small unweighting efficiency without contributing significantly to the total integral. It can therefore be useful to work with a deliberately reduced maximum, provided the rare mismatches are corrected for by event weights. The resulting events will be partially unweighted since there can be some events that overshoot the maximum. These will receive an overweight $\tilde{w} = w/w_{\max} > 1$. Hereinafter, we adopt the approach used in SHERPA for finding the reduced maximum. The aim is that the remaining overweights do not contribute more than a fixed proportion to the integral. We set this share to 0.1 %. This can be achieved by taking the sorted weights of a sample of weighted points and finding the weight that cuts off the desired quantile. In SHERPA this is done automatically during the integration phase. We point out that using a reduced maximum is a fully unbiased technique commonly used in event generators. It is especially helpful when weight surrogates are used since the limited approximation quality of the surrogate can lead to particularly large outliers.

## 3.2 Performance analysis

To fairly evaluate the performance gain of the two-stage unweighting algorithm shown in Alg. 1 we take the average time it takes to generate a single (partially) unweighted event and compare it to the time it would take to generate the statistical equivalent using the standard unweighting procedure. We call the ratio between the two the effective gain factor $f_{\mathrm{eff}}$:

$$f_{\mathrm{eff}} := \frac{T_{\mathrm{standard}}}{T_{\mathrm{surrogate}}}. \tag{11}$$

In order to separate the actual unweighting from program initialisation and other aspects of event generation, we break the calculation down to the relevant ingredients:

$$f_{\mathrm{eff}} = \frac{N_{\mathrm{full}}^{\mathrm{trials}} \cdot \left( \langle t_{\mathrm{ME}} \rangle + \langle t_{\mathrm{PS}} \rangle \right)}{N_{\mathrm{1st,surr}}^{\mathrm{trials}} \cdot \left( \langle t_{\mathrm{surr}} \rangle + \langle t_{\mathrm{PS}} \rangle \right) + N_{\mathrm{2nd,surr}}^{\mathrm{trials}} \cdot \langle t_{\mathrm{ME}} \rangle} \tag{12}$$

$$= \frac{1}{\frac{\langle t_{\mathrm{surr}} \rangle + \langle t_{\mathrm{PS}} \rangle}{\langle t_{\mathrm{ME}} \rangle + \langle t_{\mathrm{PS}} \rangle} \cdot \frac{\epsilon_{\mathrm{full}}}{\epsilon_{\mathrm{1st,surr}} \epsilon_{\mathrm{2nd,surr}}} + \frac{\langle t_{\mathrm{ME}} \rangle}{\langle t_{\mathrm{ME}} \rangle + \langle t_{\mathrm{PS}} \rangle} \cdot \frac{\epsilon_{\mathrm{full}}}{\epsilon_{\mathrm{2nd,surr}}}}. \tag{13}$$

The average evaluation times of the full matrix element weight, the phase space weight and the matrix element surrogate, respectively, are denoted as $\langle t_{\mathrm{ME}} \rangle$, $\langle t_{\mathrm{PS}} \rangle$ and $\langle t_{\mathrm{surr}} \rangle$. By $N_{\mathrm{step}}^{\mathrm{trials}}$ we denote the number of trials in the respective unweighting step. The unweighting efficiencies are defined as

$$\epsilon_{\mathrm{full}} := \frac{N}{N_{\mathrm{full}}^{\mathrm{trials}}}, \quad \epsilon_{\mathrm{1st,surr}} := \frac{N_{\mathrm{2nd,surr}}^{\mathrm{trials}}}{N_{\mathrm{1st,surr}}^{\mathrm{trials}}} \quad \text{and} \quad \epsilon_{\mathrm{2nd,surr}} := \frac{N}{N_{\mathrm{2nd,surr}}^{\mathrm{trials}}}. \tag{14}$$

It should be noted that events rejected due to phase space constraints do not affect the unweighting efficiencies since the selection cuts can be applied solely based on the kinematics without having to evaluate the matrix element.

From Eq. (13) it is clear that an important requirement for significant gains are short evaluation times for the surrogate in comparison to the full matrix element, *i.e.* $\langle t_{\text{surr}} \rangle \ll \langle t_{\text{ME}} \rangle$. Furthermore, even with a fast and accurate surrogate gains are only possible when the original unweighting efficiency $\epsilon_{\text{full}}$ is small enough. Therefore, the surrogate unweighting method is of limited use when the sampling density is very well adapted to the target. For suitable processes it will thus be important to find a good balance between fast evaluation and high accuracy of the surrogate.

The efficiency $\epsilon_{\text{full}}$ can be estimated by

$$\epsilon_{\text{full}} \approx \frac{\langle w \rangle}{w_{\text{max}}}, \tag{15}$$

from the weights $w$ generated during an initial integration run, *i.e.* after adapting the phase space generator. For $w_{\text{max}}$ we use the reduced value as described in Sec. 3.1. Analogously, we estimate $\epsilon_{\text{1st,surr}}$ by

$$\epsilon_{\text{1st,surr}} \approx \frac{\langle s \rangle}{w_{\text{max}}}, \tag{16}$$

using the same weight maximum and the surrogate weights $s$ determined for the events in the test dataset. Since the deviations of the surrogate should average out, one can expect the values of $\epsilon_{\text{full}}$ and $\epsilon_{\text{1st,surr}}$ to be close. The second unweighting efficiency $\epsilon_{\text{2nd,surr}}$ can be estimated by

$$\epsilon_{\text{2nd,surr}} \approx \frac{\langle x \rangle}{x_{\text{max}}}, \tag{17}$$

using the values $x = w/s$ determined for the events in the test dataset. The reduced maximum $x_{\text{max}}$ can be calculated analogously to $w_{\text{max}}$ with the restriction that we have to weight the values of $x$ by their corresponding values of $s$ to take into account the acceptance probability in the first unweighting step.

To determine the times $\langle t_{\text{ME}} \rangle$, $\langle t_{\text{PS}} \rangle$ and $\langle t_{\text{surr}} \rangle$ we repeat the calculation of the full/surrogate matrix element and phase space weights for a number of events from the test dataset. Depending on the complexity of the process we need between 10 and 10 000 events for a reliable time estimate. Note, the value of $\langle t_{\text{surr}} \rangle$ includes the time for preprocessing the inputs and post-processing the outputs of the surrogate model.

# 4 Implementation and application to LHC processes

In this section we present the application of the dipole model emulation of QCD matrix elements in the unweighting of event samples for high-multiplicity scattering processes at the LHC, *i.e.* $Z + 4, 5$ jets and $t\bar{t} + 3, 4$ jets production at the LHC. Results presented in Ref. [2] were based on a simplified neural network surrogate, however, also included an approximation for the phase space weight. We will here contrast the results obtained before to the sophisticated dipole model surrogate and also comment on the challenges when using colour-sampled QCD amplitudes. We furthermore briefly describe an implementation in the workflow of the SHERPA framework [20, 21]. Note that we here only need to consider the generation of the hard process partons, as this is factorised from the generation of initial- and final-state parton showers as well as non-perturbative phases such as hadronisation and the underlying event [6]. Furthermore we note that systematic variations of the hard event related to alternative PDF sets, or modifications in the scale choices can be evaluated on-the-fly for unweighted events, represented by variational weights, see for example [39, 40].

## 4.1 Implementation in the SHERPA framework

The two-stage unweighting algorithm described in Sec. 3.1 has been implemented in SHERPA [2]. The framework provides two built-in tree-level matrix element generators: AMEGIC [41] and COMIX [31]. We use AMEGIC to evaluate colour-summed matrix elements and COMIX for colour-sampled ones. To adapt the integrator to the integrand SHERPA runs an initial optimisation phase. This is followed by an integration phase in which the optimised integrator is used to calculate the total cross section of the process. From the event weights produced in this phase the value of $w_{\max}$ is determined, based on the 0.1 % maximum reduction method introduced in Sec. 3.1. We take 2M events from the integration phase as a training dataset by saving the momenta, matrix element and phase space weights, and, when using colour sampling, colour assignments. From the training dataset we use 800k events for training the model, 200k for validation during the training and 1M for testing the performance afterwards. We train the dipole model described in Sec. 2 using KERAS [24] with the TENSORFLOW [25] backend and save it in the ONNX format [42]. The 1M events from the test dataset are used to determine the value of $x_{\max}$.

After the training of the surrogate model has been completed successfully, the determination of the surrogate matrix element value during event generation with SHERPA proceeds as follows. At the point where normally the matrix element would be calculated with AMEGIC or COMIX, we use the momenta of the current trial event to determine the additional inputs $y_{ijk}$ and $s_{ij}$. Along with the momenta these are then fed into the model which we evaluate on a single CPU core using the C++ API of the ONNX Runtime package [30]. We find that ONNX Runtime evaluates the model several times faster than the header-only library frugally-deep [43], which was used in [2]. It is important to note that this introduces an additional dependency on a software library. We would, however like to emphasise that our method does not depend on the code with which the surrogate is evaluated. This affects only the evaluation time. It would even be possible to create an interface through which any suitable tool could be used for this purpose. The model evaluation yields the dipole coefficients $C_{ijk}$ which are then combined with the dipole functions $D_{ijk}$ according to Eq. (2). To determine the relevant dipoles we use a custom implementation, although there already exists an implementation of the dipole functions in SHERPA (used in the automated construction of infrared subtraction terms for NLO QCD and EW calculations [44,45]) which could in principle also be employed for the case considered here.

## 4.2 Results for LHC multijet production processes

To study the performance of the method we consider various partonic multijet processes at tree-level accuracy. We thereby follow the validation and benchmark strategies outlined in Ref. [2], considering $Z$+jets and $t\bar{t}$+jets production in proton–proton collisions at $\sqrt{s} = 13$ TeV. In particular we present results for $Z + \{4, 5\}$ jets and $t\bar{t} + \{3, 4\}$ jets final states, thereby extending our previous study by one multiplicity. Jets get reconstructed with the anti-$k_t$ algorithm [46] with $R = 0.4$. As parton density functions we use the NNPDF-3.0 NNLO set [47].

### $Z$+jets

We examine the partonic channels $gg \to e^-e^+ggd\bar{d}$ and $gg \to e^-e^+gggd\bar{d}$ at the tree-level that represent leading-order contributions to $Z + 4$ jets and $Z + 5$ jets production at the LHC. Correspondingly, using the four-momenta as inputs for the surrogate model we have parameter spaces with 32 and 36 dimensions. These get supplemented by the corresponding dipole mapping variables and kinematic invariants, see Sec. 2.2. Cuts are implemented to constrain the fiducial phase space and, in turn, to regulate QCD infrared divergences. A dilepton in-

Table 2: Performance measures for partonic channels contributing to $Z + \{4, 5\}$ jets production at the LHC.

| | SHERPA default | | | with dipole-model surrogate | | | | |
|---|---|---|---|---|---|---|---|---|
| Process | $t_{\text{ME}}$[ms] | $t_{\text{PS}}$[ms] | $\epsilon_{\text{full}}$ | $t_{\text{surr}}$[ms] | $x_{\text{max}}$ | $\epsilon_{\text{1st,surr}}$ | $\epsilon_{\text{2nd,surr}}$ | $f_{\text{eff}}$ |
| $gg \to e^- e^+ gg d\bar{d}$ | 54 | 0.40 | 1.411 % | 0.14 | 2.6 | 1.418 % | 39 % | 16 |
| $gg \to e^- e^+ ggg d\bar{d}$ | 16 216 | 5.70 | 0.076 % | 0.20 | 3.6 | 0.085 % | 29 % | 269 |

variant mass $m_{e^- e^+} > 66\,\text{GeV}$ and four, respectively, five jets with $p_{T,j} > 20\,\text{GeV}$ are enforced. Identical cuts are used for the training and the prediction.

As a first assessment of the quality of the surrogate we show in Fig. 7a the distribution of the ratio between the true event weight $w$ and the surrogate event weight $s$ for 1M test events for the exemplar channel $gg \to e^- e^+ ggg d\bar{d}$. The corresponding plot for the process with the lower multiplicity is shown in App. A. We compare the results of the dipole model with the naive model from Ref. [2]. We point out that the naive model learns the entire event weight, while the dipole model learns only the matrix element weight. For the representation in Fig. 7a, the approximated matrix element weight of the dipole model was therefore multiplied by the true phase space weight. While a perfect model would reproduce the true weight exactly, such that the ratio would be one for all events, our surrogates show deviations. In both cases the distribution is peaked at one and falls off rather symmetrically towards higher and lower values. For the dipole model the peak is more pronounced and has a steep slope towards the tails of the distribution. This indicates that for the bulk of the events the dipole model produces results that are much closer to the true values than the ones from the naive model. While the naive model seems to tend to generate an excessive number of large weights, *i.e.* $s > w$, both models generate a small number of outliers with $s \ll w$, reaching values for $x = w/s$ of up to $10^7$. We also indicate the points where the values of $x_{\text{max}}$ lie to show which parts of the distributions are cut off in the partial unweighting. The dipole model achieves a much smaller $x_{\text{max}}$ than the naive model, 3.6 compared to 84.8. There are orders of magnitude between the large-$x$ outliers and the value of $x_{\text{max}}$ used for the unweighting. To underline that this does not contradict each other, we refer again to Figs. 4–6. There we can see that the large ratios $x$ are suppressed with respect to the matrix element weight. These events therefore have a low probability of being accepted in the unweighting. Accordingly, the cut by the reduced $x_{\text{max}}$ is justifiable because the outliers on average contribute little to the total cross-section due to their low frequency.

In Tab. 2 we summarise the evaluation times of the full and dipole-model surrogate weights, the efficiencies of the single- and two-stage unweighting, the maximum $x_{\text{max}}$ for the second unweighting step and, finally, the effective gain factor $f_{\text{eff}}$. The evaluation of the surrogate is found to be orders of magnitude faster than the full matrix element calculation with AMEGIC. In the 4-jet case it is more than 300, and in the 5-jet case more than 80.000 times as fast. The evaluation of the phase space weights is fast in comparison to the full matrix element. However, it is of order, or even larger than $\langle t_{\text{surr}} \rangle$. We find that the additional complexity when increasing the multiplicity from four to five jets increases the matrix element evaluation time by a factor of 300 and reduces the unweighting efficiency by a factor of 20. Nevertheless, $\langle t_{\text{surr}} \rangle$ grows only by a factor less than two, while the approximation accuracy, reflected by $x_{\text{max}}$ and $\epsilon_{\text{2nd,surr}}$, remains very similar. We obtain the values $x_{\text{max}} = 2.6$ and $\epsilon_{\text{2nd,surr}} = 0.39$ in the 4-jet case compared to $x_{\text{max}} = 3.6$ and $\epsilon_{\text{2nd,surr}} = 0.29$ for five jets. The effective gain factors yield 16 and 269, respectively.

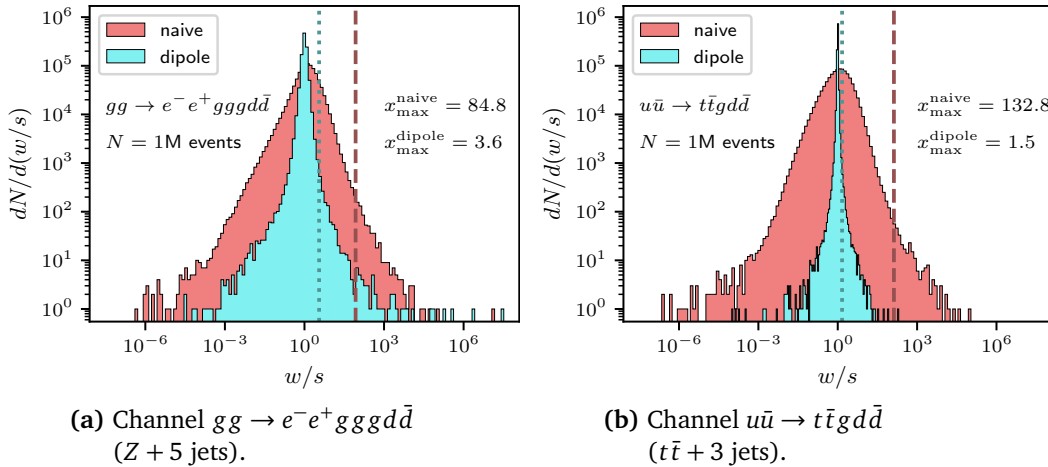

**(a)** Channel $gg \rightarrow e^- e^+ gggd\bar{d}$
($Z + 5$ jets).

**(b)** Channel $u\bar{u} \rightarrow t\bar{t}gd\bar{d}$
($t\bar{t} + 3$ jets).

Figure 7: Ratio distributions of exact weights and their surrogate for the factorisation-aware emulation of the matrix-element weight (dipole) and the combined matrix-element and phase-space weight from Ref. [2] (naive).

## $t\bar{t}$+jets

As contributions to the processes $t\bar{t} + 3$ jets and $t\bar{t} + 4$ jets in hadronic collisions we here consider three partonic channels with varying number of external gluons, namely $u\bar{u} \rightarrow t\bar{t}gd\bar{d}$, $gg \rightarrow t\bar{t}ggg$ and $ug \rightarrow t\bar{t}gggu$. In contrast to the previous examples these are pure QCD processes featuring massive coloured particles. Even though the final states contain one particle fewer than the $Z$+jets channels, these processes still pose a severe computational challenge. The direct coupling of gluons to the top quarks leads to a significant proliferation of Feynman diagrams in their jet-associated production. The input space dimensionalities are now 28 and 32, respectively. For the processes contributing to $t\bar{t} + 3$ jets we require three anti-$k_t$ jets with $p_{T,j} > 20$ GeV. The fiducial phase space of the $t\bar{t} + 4$ jets channel is constrained by requiring four jets with staggered transverse-momentum cuts, namely $p_{T,1} > 100$ GeV, $p_{T,2} > 50$ GeV, $p_{T,3} > 40$ GeV and $p_{T,4} > 20$ GeV. We do not impose phase space restrictions on the external top quarks, that we treat as on-shell in the matrix element calculation, *i.e.* $p_t^2 = p_{\bar{t}}^2 = m_t^2$ with $m_t = 173.4$ GeV.

In Fig. 7b we show the ratio distributions of the true event weights and their surrogates for the dipole model and the naive model using the example of the partonic channel $u\bar{u} \rightarrow t\bar{t}gd\bar{d}$. Note that the corresponding distributions for the other channels are shown in App. A. In comparison to Fig. 7a it can be seen that the distribution of the naive model is wider while the one of the dipole model is even narrower in this example. Moreover, it has visibly fewer outliers. This is also reflected in the values of $x_{\max}$, where the excellent result of 1.5 for the dipole model is two orders of magnitude smaller than the one for the naive model.

In Tab. 3 we compile the results obtained for the three partonic channels comparing the ordinary unweighting procedure with the two-stage surrogate technique. Again, we find significant speedups when using the dipole-model surrogate. For the process $ug \rightarrow t\bar{t}gggu$ the surrogate is in fact more than 200.000 times faster than the full matrix element weight evaluation. For all three examples the surrogate gives accurate approximations leading to values of $x_{\max}$ between 1.4 and 1.8. The gain factors $f_{\text{eff}}$ lie between 20 for the process $u\bar{u} \rightarrow t\bar{t}gd\bar{d}$ and 354 for $ug \rightarrow t\bar{t}gggu$.

We compare the results for the effective gain factors for all five example processes in Fig. 8. For comparison we also include the results obtained using the simpler NN surrogate from Ref. [2] that were not contained in the tables. The values differ from the original publication because the definition of the renormalisation and factorisation scales has changed from a

Table 3: Performance measures for partonic channels contributing to $t\bar{t} + \{3, 4\}$ jets production at the LHC.

| Process | SHERPA default | | | with dipole-model surrogate | | | | |
|---|---|---|---|---|---|---|---|---|
| | $t_{\text{ME}}$[ms] | $t_{\text{PS}}$[ms] | $\epsilon_{\text{full}}$ | $t_{\text{surr}}$[ms] | $x_{\text{max}}$ | $\epsilon_{\text{1st,surr}}$ | $\epsilon_{\text{2nd,surr}}$ | $f_{\text{eff}}$ |
| $u\bar{u} \to t\bar{t}gd\bar{d}$ | 5 | 0.04 | 0.092 % | 0.14 | 1.5 | 0.092 % | 69 % | 20 |
| $gg \to t\bar{t}ggg$ | 3262 | 0.90 | 1.093 % | 0.18 | 1.4 | 1.128 % | 69 % | 61 |
| $ug \to t\bar{t}gggu$ | 51 200 | 4.00 | 0.153 % | 0.24 | 1.8 | 0.160 % | 57 % | 354 |

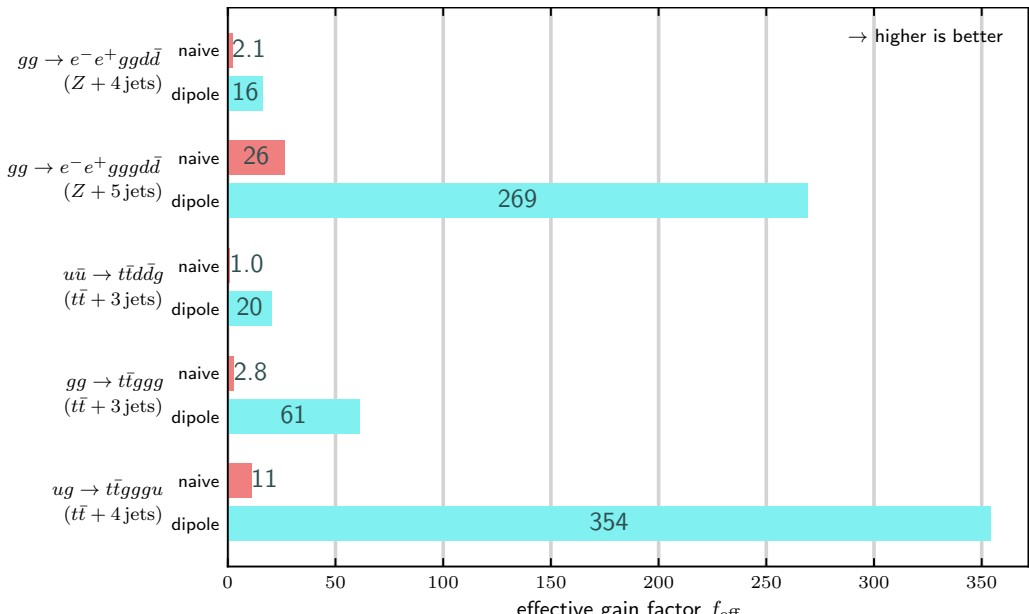

Figure 8: Effective gain factors for different processes. For comparison the results obtained using the *naive* neural network surrogate model from Ref. [2] are shown. Note that the naive model includes the phase space weight while the dipole model learns the matrix element weight only.

momenta-dependent one as used in Ref. [2] to a fixed value as used in Ref. [1]. This change leads to a slightly simpler learning problem and thus to slightly better performance. It can be seen that the dipole model achieves much larger gain factors. This can be attributed to the fact that the dipole model approximates the matrix elements much better because it already knows the relevant dipole structures for QCD emissions that dominate the multijet processes considered here. Furthermore, it is found that the respective highest multiplicity channels of the two process groups yield the largest gain factors. Adding an additional external particle causes the complexity of the calculation of the matrix element to grow significantly. This leads to a considerably increased evaluation time $t_{\text{ME}}$ for the full weight, while the time $t_{\text{surr}}$ for the surrogate changes only insignificantly. The impressive performance of the dipole surrogate model facilitates high gains even for those channels where the naive model from Ref. [2] led to minor gains only.

**The influence of the training dataset size**

In Fig. 9 we show how the value of $x_{\text{max}}$ depends on the event sample size used to train the surrogate model for the different example processes. The number of training events is varied

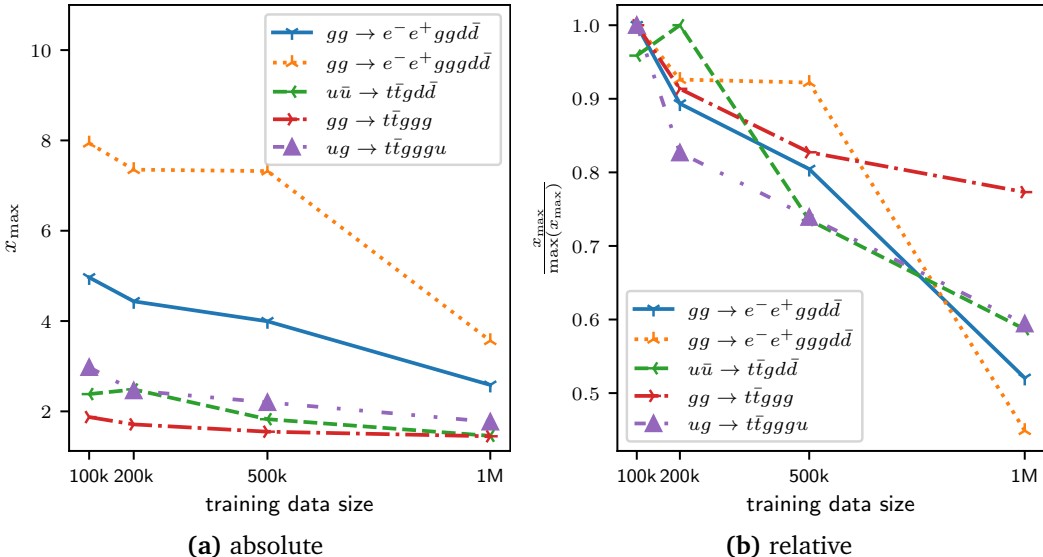

**(a)** absolute  **(b)** relative

Figure 9: Influence of the training data size on the value of $x_{\max}$.

between $10^5$ and $10^6$. A hierarchy can be identified: the models with the highest, *i.e.* worst, values of $x_{\max}$ gain the most from additional training data. For the process $gg \to e^- e^+ g g g d \bar{d}$ for example the resulting $x_{\max}$ is more than halved by going from $10^5$ to $10^6$ events. The processes with smaller $x_{\max}$ in comparison benefit less. For the process $gg \to t \bar{t} g g g$ the gain is only 23 %. These observations carry over to Fig. 10 where the dependence of $f_{\mathrm{eff}}$ on the training dataset size is shown. According to Eq. (13) we have $f_{\mathrm{eff}} \propto \epsilon_{\mathrm{2nd,surr}}$ and according to Eq. (15) we have $\epsilon_{\mathrm{2nd,surr}} \propto 1/x_{\max}$. Therefore $f_{\mathrm{eff}}$ is inversely proportional to $x_{\max}$. The largest improvement can again be seen for the process $gg \to e^- e^+ g g g d \bar{d}$ where the value of $f_{\mathrm{eff}}$ increases by 125 % when going from $10^5$ to $10^6$ events. Likewise, the smallest improvement relates to the process $gg \to t \bar{t} g g g$ where the increase is only 22 %.

**Results for colour-sampled amplitudes**

The above examples are based on matrix elements with an explicit sum over the $SU(3)$ colour configurations of the involved partons. Using Monte Carlo integration techniques for phase space sampling, and possibly partonic flavours, a further option arises: just like the kinematic variables, we can also sample the colour assignments for the external partons. It can be shown [48] that colour sampling has a superior scaling behaviour compared to colour summation and therefore becomes much faster for large parton multiplicities. This holds even though colour sampling needs more points to reach a certain target precision. With $6 - 8$ colour charged legs our examples already feature quite high dimensional colour spaces. It is thus worthwhile to test the performance of our method for colour sampled matrix elements. As a benchmark we use SHERPA with its built-in matrix element generator COMIX that implements colour sampling based on the colour-flow decomposition of QCD amplitudes [31,32]. To keep things simple, we use a naive approach and employ basically the same surrogate model as before with the colour configuration as an additional input, see Sec. 2.3. While our model ansatz Eq. (2) is averaged over the colours, the neural network can try to learn the colour structure and encode it in the coefficients. As discussed in Sec. 2.3, an improved approach could use a new set of dipoles with explicit colour assignment in the future.

We trained the dipole model on the processes $gg \to e^- e^+ g g d \bar{d}$ and $gg \to t \bar{t} g g g$ and found gain factors of 0.23 and 0.26, respectively. The performance is thus worse than using the standard unweighting when sampling colours. We checked that increasing the size of

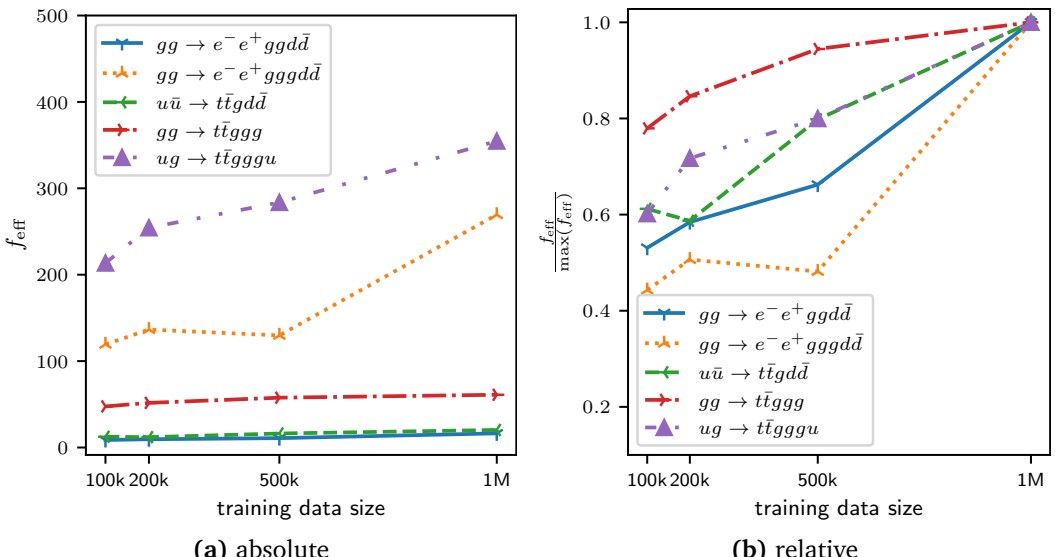

**(a)** absolute $\qquad\qquad\qquad\qquad$ **(b)** relative

Figure 10: Influence of the training data size on the value of $f_{\text{eff}}$.

the training dataset does not lead to much higher gains. Three effects come into play here: first, the approximation quality of the model is worse because the complexity of the emulation problem increases significantly due to the additional colour degrees of freedom. Secondly, the evaluation time $t_{\text{ME}}$ for the matrix element is now much shorter because instead of the whole sum only a single colour point needs to be evaluated. Thirdly, the evaluation time $t_{\text{PS}}$ for the phase space weight is now no longer negligible. With COMIX it is of the same order of magnitude as $t_{\text{ME}}$. This makes it much more difficult to achieve large gains.

A way to deal with the last two points would be to let the surrogate also approximate the phase space weight such that

$$s' \approx w_{\text{ME}} \cdot w_{\text{PS}} \,. \tag{18}$$

Let us demonstrate this for the effective gain factor. In the limit of a highly accurate surrogate with $\epsilon_{\text{1st,surr}} \approx \epsilon_{\text{full}}$ and $\epsilon_{\text{2nd,surr}} \approx 1$ Eq. (13) becomes:

$$f_{\text{eff}} \approx \frac{1}{\frac{\langle t_{\text{surr}} \rangle + \langle t_{\text{PS}} \rangle}{\langle t_{\text{full}} \rangle} + \frac{\langle t_{\text{ME}} \rangle}{\langle t_{\text{full}} \rangle} \cdot \epsilon_{\text{full}}} \,. \tag{19}$$

Even in the ideal case where $\langle t_{\text{surr}} \rangle \to 0$ and $\epsilon_{\text{full}} \to 0$ there is an upper limit given by

$$f_{\text{eff}} \leq \frac{\langle t_{\text{full}} \rangle}{\langle t_{\text{PS}} \rangle} \,. \tag{20}$$

This is unproblematic as long as the evaluation of $w_{\text{PS}}$ is cheap compared to $w_{\text{ME}}$. If this is not the case a surrogate that emulates the full weight is beneficial and results in an effective gain factor of:

$$f'_{\text{eff}} = \frac{1}{\frac{\langle t'_{\text{surr}} \rangle}{\langle t_{\text{full}} \rangle} \cdot \frac{\epsilon_{\text{full}}}{\epsilon'_{\text{1st,surr}} \epsilon'_{\text{2nd,surr}}} + \frac{\epsilon_{\text{full}}}{\epsilon'_{\text{2nd,surr}}}} \,. \tag{21}$$

Considering again the limit of a highly accurate surrogate leads to

$$f'_{\text{eff}} \approx \frac{1}{\frac{\langle t'_{\text{surr}} \rangle}{\langle t_{\text{full}} \rangle} + \epsilon_{\text{full}}} \,. \tag{22}$$

The largest possible gain factor is thus $f'^{\text{max}}_{\text{eff}} = \epsilon_{\text{full}}^{-1}$. This corresponds to the same acceptance rate as without surrogate but with zero evaluation time. As was done in Ref. [2] we adapted

the dipole surrogate model to include the phase space weight and evaluated the performance for the same two processes as above. We find gain factors of 0.02 and 0.22, respectively. Again, we do not achieve any gains compared to the standard unweighting. In this case the problem is that the neural network gives an even worse approximation since we include the phase space mapping which already tries to flatten the structures in the soft and collinear regions. So the model has to deal with a situation it was originally not designed for. The resulting losses eat up the gain from not having to calculate $w_{\text{PS}}$ for every trial event.

The observations described above open up various options for improvement. One possibility, as mentioned before, would be to develop a surrogate model with colour-dependent dipoles, adequately representing amplitudes in a specific colour-flow assignment. In addition, one could attempt to explicitly incorporate knowledge about the employed phase space mappings.

# 5 Conclusions

We presented a case study of using a fast and accurate neural network emulation model for scattering matrix elements in the context of unweighted event generation for multijet processes. To this end we have generalised the model originally presented in Ref. [1], based on dipole factorisation, to account also for initial-state emissions and massive final-state partons. When considering QCD multijet processes this factorisation-aware model – using the parton four-momenta, dipole variables and kinematic invariants as inputs – provides very precise estimates for the squared transition amplitudes. This has been showcased for a selection of partonic channels contributing at the tree-level to hadronic $Z + 4, 5$ jets and $t\bar{t} + 3, 4$ jets production.

We then considered the trained networks in the ONNX format as fast surrogates for the full squared matrix elements in a two-stage rejection algorithm, originally presented in Ref. [2], in the SHERPA framework. This enables the production of unbiased samples of unweighted events that reproduce the exact target distribution, *i.e.* the true squared matrix element of the considered scattering process. Given a fast *and* accurate surrogate model, the effective gains are largest when two conditions are met: (i) the unweighting efficiency of the phase space integrator is rather low, and, (ii) the matrix element is time consuming to evaluate. For example, for the channels $gg \rightarrow e^-e^+ggg d\bar{d}$ and $ug \rightarrow t\bar{t}gggu$ in proton–proton collisions at $\sqrt{s} = 13$ TeV, featuring default unweighting efficiencies for the AMEGIC integrator of 0.08% and 0.153%, we found gain factors of 269 and 354, respectively, when using our dipole-model surrogate. Accordingly, the computational resources needed to generate a given number of unweighted events get reduced by more than two orders of magnitude. At the same time, the overheads for training the surrogate network model are very modest, given that events from the compulsory integration phase prior to the generation process can be used for that purpose.

The underlying workflow for colour-summed squared matrix elements should be easily adaptable also for other matrix element providers and usage in experimental computing frameworks, given that in contrast to the original treatment from Ref. [2] we only employ the emulation of the matrix element expression and no longer include the generator specific phase space weight in the first-stage approximation. Furthermore, the ONNX standard allows one to easily store, transfer and exchange the trained neural networks, offering much flexibility in the method used to train the model.

Our results are valid for a sequential event generation workflow where events are generated one after the other on a single CPU core. We expect that the performance can be further increased by moving to a parallel workflow that generates multiple events at the same time using parallel hardware. The evaluation of the neural networks, which form the basis for the

surrogate models, can be easily vectorised and benefits in particular from accelerators such as GPUs.

Our study targeted high-multiplicity tree-level contributions that constitute a severe computational challenge in state-of-the-art matrix element plus parton shower simulations of multijet production processes [49–54], given that one can typically achieve NLO QCD accuracy only for somewhat lower multiplicities, see for instance [55]. However, in particular for the highest multiplicities sampling the colour assignments of the external partons outperforms their explicit summation. This poses new challenges to emulation models, given the high-dimensionality of the colour space. We explored naive extensions towards a suitably adjusted network model, though we were not able to achieve significant gains using a surrogate based on colour-summed dipoles. This is partly also due to the reduced evaluation times for partial amplitudes in the colour-flow decomposition. We are confident that under the same strategy but using colour-stripped dipoles in the surrogate ansatz and incorporating the phase space weight into the emulation useful gain factors could be achieved.

While the current paper addressed the emulation of high-multiplicity tree-level matrix elements, an extension to NLO calculations is rather straightforward, though there are additional challenges that have to be addressed. In Ref. [2] it was already shown that the two-staged unweighting algorithm is applicable also to non-positive definite target functions, *i.e.* negative event weights as they appear in subtraction-based higher-order calculations. In Ref. [56] an extended version of the factorisation-aware emulation model used here to QCD one-loop matrix elements has been presented. For the considered multi-jet production channels in electron–positron annihilation percent-level accuracy has been achieved. Upon generalisation to initial-state partons this could be used to lift unweighted event generation for LHC applications based on fast neural-network surrogates to NLO accuracy.

# Acknowledgements

We are grateful for fruitful discussions with Enrico Bothmann, Stefan Höche, and Max Knobbe.

**Funding information** The work of SS and TJ was supported by BMBF (contract 05H21MGCAB) and Deutsche Forschungsgemeinschaft (DFG, German Research Foundation) - project number 456104544. FS's research was supported by the German Research Foundation (DFG) under grant No. SI 2009/1-1. DM's research was supported by STFC under grant ST/X003167/1.

# A   Auxiliary weight distributions

In this appendix we collect in Figs. 11a–11c auxiliary plots for the emulation accuracy of our dipole surrogate model (dipole) and the combined neural network surrogate for the matrix-element and phase-space weight from Ref. [2] (naive) for the remaining partonic channels. Shown are the ratios of the true weights and the respective surrogates. The two vertical lines indicate the corresponding maxima based on the 0.1 % maximum reduction method, see Sec. 3.1.

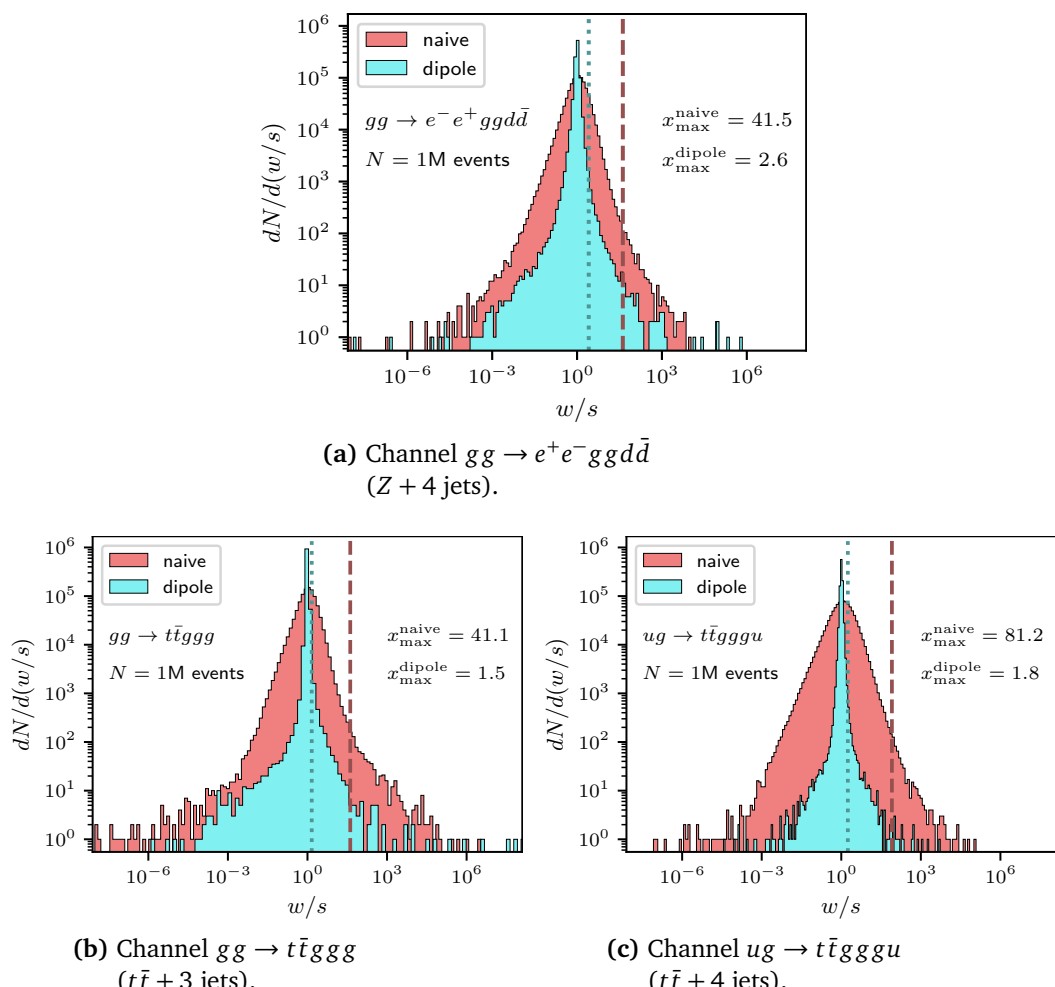

**(a)** Channel $gg \to e^+e^- ggd\bar{d}$ ($Z + 4$ jets).

**(b)** Channel $gg \to t\bar{t}ggg$ ($t\bar{t} + 3$ jets).

**(c)** Channel $ug \to t\bar{t}gggu$ ($t\bar{t} + 4$ jets).

Figure 11: Ratio distributions of exact weights and their surrogate for the factorisation-aware emulation of the matrix-element weight (dipole) and the combined matrix-element and phase-space weight from Ref. [2] (naive) for different partonic channels.

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
