# Peer review of "Unweighting multijet event generation using factorisation-aware neural networks"

_SciPost Physics, doi:SciPost Phys. 15, 107 (2023)_

## Round 1 · Referee Report · Anonymous (Referee 1) · 2023-3-1

Report

This is an interesting article, addressing a well-known problem, that deserves publishing. I have no major physics objections. Not being an expert, I have been confused in some places, however, and then others may be as well. Therefore I would ask the authors to seriously consider whether a more pedagogical presentation is possible for the points raised below.

1) Normally one does not include references in the abstract, but only in the main body of the event. Could this be respected here as well, without any lack of readability?

2) Eq. (1) defines a recursive procedure, where an n+1 configuration can be approximated by n, and in turn by n-1, n-2, and so on down to a minimal topology. But eq. (2) would seem to indicate that only the first step in this procedure will be approximated, while the C_ijk coefficients will contain the full complexity of the n-parton matrix elements (MEs). It would be useful to clarify this to the reader. Take e.g. the double-colliner limit, where three partons are almost collinear. Is it then to be assumed that one singularity will be handled by the predetermined D_ijk factors, while the C_ijk coefficients will have to reproduce the other singularity factor, whithout any prior knowledge about it? Or have I misunderstood? If so, all the more reason to write it out clearly.

3) For those of us not NN experts, it would have been helpful to write out the shape of the swish activation function. Now I had to look it up, only to find that it is as simple as x/(1 + exp(-x)), which is so short an expression that you can easily have it inline in the text, and then its listed properties become obvious immediately. Maybe there is also a simple way to be a bit more explicit what Glorot is all about.

4) It could be useful and would not take much space to show what an y_ijk expression looks like, at least for the massless FF case. This would then naturally lead up to eq. (3).

5) In view of the many input and output transformations used, it would be helpful to be more explicit what is meant by loss in Fig. 2. Is it the RMS of the true/predicted (squared) ME ratio, where then the vertical axis has an easily understood meaning, or is it based on transformed variables, where this is less obvious?

6) On p. 8 the different IF, FI and II dipoles are introduced, but actually that terminology is already assumed for eq. (3). Reorder.

7) The results of Fig. 4 (and similar subsequent ones) are presented in a positive spirit, while they rather make me worried. In a sample of 10^6 events, it appears that a few events have truth/predicted weights close to 10^6. a) Does that imply that each such high-weight event has as much impact on final distributions as the combined bulk of events with weight close to 1? If so, a convenient way to show this is to add a further histogram curve to the right-side projection, where each event has been filled with its respective weight rather than unity. b) If you were to extend the study to 10^7 events, is it then possible that you would find events with weights around 10^7, and so on? (in Fig. 5 weights stretch to 10^8.) Then you would have a serious instability that should be mentioned. c) What is the phase-space region actually sampled? Are results extreme because you are exploring soft/collinear regions that would be entrusted to showers in practical applications? It could then be useful to relate to the behaviour also for events with better separated partons? Notably, the narrowing of the weight distribution for the largest true ME values may look nice, but what if the experimentally interesting events are found for somewhat lower values, where the spread is larger?

8) g is introduced just above eq. (6) as a probability density, which means that implicitly its integral is normalized to unity. It would improve readability to make this explicit in eq. (6), e.g. I = \int f du = \int (f/g) dg du with \int g du = 1 (in shorthand, hopefully you get the idea). After eq. (7) you then also add that the points u_i are drawn from the g distribution, as reflected in the <w>_g notation.

9) In mid-page 14 you define 0.1% as an acceptable influence of overweight events. But this is for the full cross section integral, right? Which is dominated by high-ME events in the soft/collinear regions, cf. 7c above. Do you have any estimate how much worse than 0.1% it could get in specific regions of phase space, e.g. for better separated partons? Recall that a typical experimental application would be to plot the pT spectrum of the n+1'th jet, and then one would like this spectrum to be well predicted both at small and large pT scales, i.e. large and small cross sections.

10) On top of p. 17 a R = 0.4 is introduced as a jet clustering scale. Should this be viewed as defining the phase-space border, or is that separate? Cf. point 7c.

11) On mid-p. 17 you note that x = w/s can be as high as 10^7, which would seem to be inconsistent with an x_max = 3.6 and 0.1% allowed weight contribution above that. Time to briefly remind the reader that this is only consistent because 10^7 happens in events with small ME weight? Cf. point 9.

12) In summary, impressive improvements. For the future, two questions remain in my mind. a) Could you break out further layers of dipole factors in eq. (2)? (Point 2.) b) The extremely large weights that can occur are worrisome (point 7). Would it be possible and useful to increase the penalty for large weights (and decrease it for small ones) in the training of the network?

---

## Round 1 · Referee Report · Anonymous (Referee 2) · 2023-4-10

Strengths

1- Developed a non-trivial efficient and reliable NN model for the emulation of (high-multiplicity) matrix elements applicable to massless and massive processes at hadron colliders. This as an extension from earlier work by some of the authors which was then applied to massless processes at e+e- colliders

2- Usage of the earlier emulation to produce a two-step unweighting algorithm based on a proposal earlier presented by some of the authors

3- Obtained very impressive speed up of the production of unweighted events for complex processes like Z+4,5 jets and ttbar+3,4 jets

Weaknesses

1- Emulation of the matrix elements was performed including color sums for colored particles, which is known to be outperformed by Monte Carlo samplings. Nevertheless, the authors already consider an application without the color sums, and although not very successful it opens ways for future improvements

Report

The future of collider phenomenology will hinge on our ability to simulate the enormous amount of data that will be produced for example at the HL-LHC. It is well known that a key bottleneck in such task is the fast simulation of complex unweighted events coming from the hard interaction. By complexity we mean processes that involve many colored partons in the final state.

In this article the authors target head on these difficulties by combining and further developing two methods proposed earlier (and separately) by some of the authors. Those are 1) a factorization-aware matrix-element emulation, and 2) a two-step unweighting procedure based on a fast surrogate for the computation of corresponding matrix elements and phase-space weights

Further developments that are carried in this article include: extension of factorization-aware emulation to processes that involve initial-state partons (to be applied to processes at hadron colliders); including radiation dipoles when massive partons are present; decoupling of surrogate calculation of matrix elements and phase-space weights.

The authors show impressive results, in particular when producing unweighted events for Z+5-jet production and ttbar+4-jet production, where more than two-orders of magnitude are gained in computational speed.

Further improvements are left for future developments, including a better emulation of matrix elements without color sums and a better emulation of the computation of phase-space weights.

Given the novel developments and promising results shown, which are of key interested for the future program at the HL-LHC, I recommend the article for publication.

Requested changes

1) The authors call systematically the radiation dipoles the "Catani-Seymour dipoles". But, as cited in the text, the massive dipoles are due to Catani, Dittmaier, Seymour, and Trocsanyi. Given the important results shown involving processes with a ttbar pair, I would recommend the authors change their terminology, in the cases when they refer collectively to massless/massive dipoles.

2) I'm confused by the multiple choices in equation (3). I think two choices suffice. One "if massless FI, IF, or II dipole", and second "otherwise" (with no parenthetical comment). This in particular more clearly state the choice for FF and FI massive dipoles.

3) The authors mention that lower-multiplicity processes (e.g. Z+2, 3 jets or ttbar+1,2 jets) can't be outperformed by their methodology, mostly due to the fast evaluation of corresponding tree-level matrix elements. Nowadays those types of simpler processes are simulated including (at least) NLO QCD corrections. The evaluation of corresponding matrix elements is much more complex as compared to the tree-level case. Can the authors comment, for example in the conclusions, what kind of extensions they would need to also emulate these more complex matrix elements?

---

## Round 2 · Author Response

Dear Editor,

We have revised our manuscript with the requested changes. Our thanks go to the referees for their helpful comments, which improve the correctness and readability of the article. We hope that the manuscript can be published in its current form.

All the best,

the authors

---

## Round 2 · List of Changes

see replies to the referees

---

## Editorial Decision

published